# Ancient mitochondrial DNA pathogenic variants putatively associated with mitochondrial disease

Draga Toncheva[1,2]*, Dimitar Serbezov[1], Sena Karachanak-Yankova[1,3], Desislava Nesheva[1]

**1** Department of Medical Genetics, Medical University of Sofia, Bulgarian Academy of Science, Sofia, Bulgaria, **2** Bulgarian Academy of Sciences–BAS, Sofia, Bulgaria, **3** Department of Genetics, Faculty of biology, Sofia University "St. Kliment Ohridski", Sofia, Bulgaria

* dragatoncheva@gmail.com

**Data Availability Statement:** All relevant data are within the paper.

**Funding:** The authors received no specific funding for this work.

## Abstract

Mitochondrial DNA variants associated with diseases are widely studied in contemporary populations, but their prevalence has not yet been investigated in ancient populations. The publicly available AmtDB database contains 1443 ancient mtDNA Eurasian genomes from different periods. The objective of this study was to use this data to establish the presence of pathogenic mtDNA variants putatively associated with mitochondrial diseases in ancient populations. The clinical significance, pathogenicity prediction and contemporary frequency of mtDNA variants were determined using online platforms. The analyzed ancient mtDNAs contain six variants designated as being "confirmed pathogenic" in modern patients. The oldest of these, m.7510T>C in the *MT-TS1* gene, was found in a sample from the Neolithic period, dated 5800–5400 BCE. All six have well established clinical association, and their pathogenic effect is corroborated by very low population frequencies in contemporary populations. Analysis of the geographic location of the ancient samples, contemporary epidemiological trends and probable haplogroup association indicate diverse spatiotemporal dynamics of these variants. The dynamics in the prevalence and distribution is conceivably result of *de novo* mutations or human migrations and subsequent evolutionary processes. In addition, ten variants designated as possibly or likely pathogenic were found, but the clinical effect of these is not yet well established and further research is warranted. All detected mutations putatively associated with mitochondrial disease in ancient mtDNA samples are in tRNA coding genes. Most of these mutations are in a mt-tRNA type (Model 2) that is characterized by loss of D-loop/T-loop interaction. Exposing pathogenic variants in ancient human populations expands our understanding of their origin and prevalence dynamics.

## Introduction

The scarcity of prehistoric human remains hampers obtaining complete picture of disorder incidence in ancient times. Altered or affected bones in skeletal remains might provide information about certain diseases, such as cancers [1, 2] and rheumatic diseases [3]. A lesion on an

**Competing interests:** The authors have declared that no competing interests exist.

archaic *Homo* mandible from Kanam, Kenya (Middle to Late Pleistocene) [4] and a fibrous dysplasia on a Neanderthal rib (older than 120000 years) from the site of Krapina, Croatia [5] are early confirmation of neoplastic disease. Neoplastic tumors have however been detected in early *Homo* samples as old as 1.7 million years ago, and these provide further insight into the outset of human cancers [6]. Mummified human remains of a 5300-year-old Neolithic man (Ötzi, The Tyrolean Iceman) show hardening of the arteries, suggesting predisposition for coronary heart disease [7]. Based on information on subsistence, geography and sample age, Berens *et al* (2017) estimate the genetic disease risk for 3180 loci in 147 ancient genomes, and find it to be similar to that of modern day humans [8]. Focusing on individual genomes, however, they estimate that the overall genomic health of the Altai Neanderthal is worse than 97% of present day humans and that Ötzi the Tyrolean Iceman had a genetic predisposition to gastrointestinal and cardiovascular diseases [8].

Data on the prehistoric origin of mitochondrial diseases is however notably lacking. The first disease linked directly to mtDNA mutation was discovered in 1988 [9]. Recently, whole-genome sequencing of mtDNA has led to significant advances in our understanding of mitochondrial diseases. Rare pathogenic mutations in mitochondrial DNA cause monogenic mitochondrial diseases involving multiple systems and are associated with variable clinical phenotypes. The severity of the clinical and biochemical phenotype caused by pathogenic mtDNA mutations has been found to be roughly proportionate to the percent mutant heteroplasmy [10, 11]. The mitochondrial haplogroup harboring the mutation might also alter the penetrance of mitochondrial diseases [12]. Specific subclades of haplogroup J, for example, have been shown to affect the penetrance and pathogenicity of Leber's hereditary optic neuropathy [13]. Certain mtDNA mutations and haplogroups are also predictors of both lifespan and risk of various age-associated disease, including degenerative diseases, cancer, diabetes, heart failure, sarcopenia and Parkinson's disease [14].

We had previously performed whole-genome sequencing on 25 Thracian mtDNA samples dated 3000–2000 BCE, and 608 mtDNA variants were detected [15]. Only one of these however, m.15326A>G (rs2853508), is designated as likely pathogenic, associated with familial breast cancer [16]. This variant was found in all analyzed by us samples, and MitoMap (2019, update nr.3) database estimates 0.98 population frequency [17]. Such high frequency indicates that this variant is common and probably not disease related.

The objective of this study was to investigate the prevalence of pathogenic mutations in ancient mtDNA and to provide further insight into the emergence and dynamics of mitochondrial diseases.

## Materials & methods

We used the comprehensive data of the ancient mtDNA genome sequences from the Ancient mtDNA database [18] which compiles human mitochondrial variation studies in ancient populations. We analyzed the *fasta* files of 1443 samples from different periods: Paleolithic, 10 samples; Mesolithic, 96; Neolithic, 341; Copper, 242; Bronze, 368; Iron, 152 and Middle Ages, 234. The total number of unique variants detected in these samples was 3191.

We used various publicly available databases to collect information on mtDNA variants, including their clinical significance and contemporary population frequencies:

- MitoMap (2019, update nr.3) is a human mitochondrial genome database that contains 14383 SNVs, 49135 full-length sequences and 72235 control region sequences [17].

MitoMap classifies a variant as being "confirmed pathogenic" if it meets the set of criteria outlined by Mitchell et al. 2006 [19], Yarham et al. 2011 [20], Wong 2007 [21] and

Gonzalez-Viogue et al. 2014 [22]: (1) independent reports of two or more unrelated families with evidence of similar disease; (2) evolutionary conservation of the nucleotide (for RNA variants) or amino acid (for coding variants); (3) presence of heteroplasmy; (4) correlation of variant with phenotype / segregation of the mutation with the disease within a family; (5) biochemical defects in the OXPHOS genes constituent complexes I, III, or IV in affected or in multiple tissues; (6) functional studies showing differential defects segregating with the mutation (cybrid or single fiber studies); (7) histochemical evidence of a mitochondrial disorder; and (8) for fatal or severe phenotypes, the absence or extremely rare occurrence of the variant in large mtDNA sequence databases.

- MITOMASTER was used to identify, annotate and evaluate the potential biological significance of nucleotide variants [23].

- GenBank database provides access to up-to-date and exhaustive DNA sequence information, and was used to get information on variant frequencies in contemporary populations [24].

- MitoTIP is an *in silico* tool for predicting pathogenicity of novel mitochondrial tRNA variants [25]. It integrates multiple sources of information, including the position of the variant within the tRNA, conservation across species and population frequencies, to provide a prediction for the likelihood that novel single nucleotide variants would cause disease.

- HmtVar uses algorithms to determine the importance of the variant position in tRNAs and was utilized to predict the pathogenicity and potential impact of mtDNA variants [26].

- Complementing the information obtained using the abovementioned tools, literature survey was conducted on variants designated as "confirmed pathogenic|" in an effort to acquire a comprehensive picture of the evidence for their disease causing effect.

## Results

Out of 3191 unique mtDNA variants established in the 1443 analyzed ancient samples, six are designated as being "confirmed pathogenic" and 10 as "likely/possibly pathogenic" by Mito-Map (Table 1). For each of these variants, we review the available evidence in HmtVar and in the scientific literature for their pathogenic effects.

**Table 1. Mitochondrial DNA mutations established in ancient samples putatively associated with mitochondrial disease.**

| | | | | | Pathogenic mutations | | | | | |
|---|---|---|---|---|---|---|---|---|---|---|
| Nr | Variants [17, 18]/ Gene product/ Strand | Period/Years [18] | Hg | Freq % in haplogroup | Country / Location / Sample | GB Freq. | Modern populations | Mito Tip classification | HmtVar Prediction | Mitochondrial Disease (MitoMap) |
| 1 | m.5703G>A (rs199476130) tRNA Asparagine /L | Neolithic (2880–2776 BCE) | K1a1b1e | 0.0 (K1a) | Poland / Koszyce / RISE1170 [27] | 0.00% | African-American woman [28], 16-year-old Caucasian girl [29], Chinese girl [30] | Cfrm Path. | Pathogenic (M2-P27) | CPEO / MM / COX deficiency (Heteroplasmy) |
| | | Iron Age (900–600 BCE) | H5a1 +152 | 0.0 (D4b) | Russia / Grishkin Log 1 / DA4 [31] | | | | | |
| | | Iron Age (900–600 BCE) | U2e1h | 0.0 (H2a) | Russia / Grishkin Log 1 / DA8 [31] | | | | | |
| | | Iron Age (156–134 BCE) | D4b2b2b | 0.0 (D4b) | Mongolia / Omnogobi / DA45 [31] | | | | | |
| | | Middle Ages (10–375 CE) | H28a1 | 0.0 (H28a) | Poland / Kowalewko, Greater Poland / PCA0002 [32] | | | | | |
| 2 | m.3243A>G (rs199474657) tRNA Leucine/H | Bronze Age (2029–1911 BCE) | T2f8a | 1.18 (T2) | Germany / Haunstetten, Postillionstr / POST_16 [33] | 0.02% | 202 symptomatic patients, UK [34], 126 carriers of the mutation, Italy [35], US [16], Netherland [36], Denmark [37], Austria [38] | Cfrm Path. | Pathogenic (M0-P14) | MELAS / LS / DMDF / MIDD / SNHL / CPEO / MM / FSGS / ASD / Cardiac+multi-organ dysfunction (Heteroplasmy) |

*(Continued)*

**Table 1.** (Continued)

| | | | | | | | | | |
|---|---|---|---|---|---|---|---|---|---|
| | | | | **Pathogenic mutations** | | | | | |
| Nr | Variants [17, 18]/ Gene product/ Strand | Period/Years [18] | Hg | Freq % in haplogroup | Country / Location / Sample | GB Freq. | Modern populations | Mito Tip classification | HmtVar Prediction | Mitochondrial Disease (MitoMap) |
| 3 | m.5650G>A tRNA Alanine/L | Iron Age (900–600 BCE) | W1c | 0.0 (H2a) | Russia / Grishkin Log 1 / DA5 [31] | 0.00% | UK [39], Germany [40], Finland [41] | Cfrm Path. | Pathogenic (M2-P6) | Myopathy (Heteroplasmy) |
| | | Iron Age (766–729 BCE) | F1b1 +@152 | 0.0 (H2a) | Kazakhstan / Kurgan Borli, Osakarovskij / DA11 [31] | | | | | |
| | | Iron Age (156–134 BCE) | D4b2b2b | 0.0 (D4b) | Mongolia / Omnogobi / DA45 [31] | | | | | |
| 4 | m.8340G>A, tRNA Lysine/H | Middle Ages (10–375 CE) | HV18 | 0.0 (H2a) | Poland / Kowalewko, Greater Poland / PCA0018 [32] | 0.00% | Canada [42], UK [43], Denmark [44] | Cfrm Path | Pathogenic (M2-P51) | Myopathy/Exercise Intolerance/Eye disease+SNHL |
| 5 | m.14674T>C, tRNA Glutamic acid/L | Bronze Age (1592–1591 BCE) | U2e1h | 0.0 (H2a) | Mongolia / Mitjurino / DA231 [31] | 0.01% | US [16], UK [45], Germany [46], Brazil [47], Italy [47], Sweden [47], Japan [48] | Cfrm Path | Pathogenic (M2-P73) | Reversible COX deficiency myopathy (Homoplasmy) |
| 6 | m.7510T>C (rs199474820) tRNA Serine/L | Neolithic (5800–5400 BCE) | J2b1 | 0.0 (J2b) | Bulgaria / Malak Preslavets / I1109 [49] | 0.00% | UK [50], Spain [51], Japan [52], Finland [53], Hungary [54], North American Caucasian families [55] | Cfrm Path | Pathogenic (M1-P5) | SNHL (Heteroplasmy) |
| | | | | | **Likely/Possibly pathogenic** | | | | | |
| 7 | m.1624C>T tRNA Valine/H | Middle Ages (10–375 CE) | T2e | 0.0 (U3a) | Poland / Kowalewko, Greater Poland / PCA0037 (55) | 0.00% | Japanese [56] | PP | Pathogenic (M2-P25) | Leigh Syndrome (Homoplasmy) |
| 8 | m.4440G>A tRNA Methionine/H | Neolithic (2800–2776 BCE) | T2b | 0.0 (T2b) | Poland / Koszyce/ RISE1159 (55) | 0.00% | Spain [57] | PP | Pathogenic (M2-P42) | MM (Heteroplasmy) |
| | | Neolithic (2800–2776 BCE) | K1a1b1e | 0.0 (K1a) | Poland / Koszyce/ RISE1162 (55) | | | | | |
| | | Neolithic (2800–2776 BCE) | HV0a | 0.0 (V7) | Poland / Koszyce/ RISE1163 (55) | | | | | |
| | | Neolithic (2800–2776 BCE) | J1c3f | 0.0 (J1c) | Poland / Koszyce/ RISE1166 (55) | | | | | |
| | | Neolithic (2800–2776 BCE) | HV0a | 0.0 (V7) | Poland / Koszyce/ RISE1173 (55) | | | | | |
| | | Iron Age (900–600 BCE) | F1b1b | 0.0 (M) | Russia / Grishkin Log 1 / DA9 [31] | | | | | |
| | | Iron Age (800–773 BCE) | H10 | 0.0 (D4) | Kazakhstan / Birlik, Kurgan 12, Bajanaul DA17 [31] | | | | | |
| | | Iron Age (600–400 BCE) | C4a1a | 0.0 (H2a) | Kazakhstan / Karaganda, Kurgan Sjartas, Shetskiy / DA10 [31] | | | | | |
| | | Iron Age (357–342 BCE) | C4a1a +195 | 0.0 (H2a) | Kazakhstan / Kurgan Sjiderti 17, Burial 1, Sjiderte, Pavlodar / DA20 [31] | | | | | |
| | | Iron Age (201–148 BCE) | D4b2b4 | 0.0 (H2a) | Mongolia / Hovsgol, Grave #18 / DA38 [31] | | | | | |
| | | Iron Age (49 BCE-53 CE) | N9a2a | 0.0 (N9a) | Mongolia / Arkhangai, Grave #1 / DA39 [31] | | | | | |
| | | Iron Age (47 BCE-24 CE) | U4a2 | 0.0 (H2a) | Kazakhstan / Naurzum, Kurgan (3), Naurzumskijzapobednik DA30 [31] | | | | | |
| | | Iron Age (177–190 CE) | A16 | 0.0 (A16) | Kazakhstan / Kurgan nr. 50, Japyryk / DA81 [31] | | | | | |
| | | Iron Age (397–570 CE) | H13a2a | 0.0 (H13a) | Kazakhstan / Kurgan nr. 1, Baskya 2 / DA385 [31] | | | | | |
| | | Middle Ages (550–850 CE) | A15c | 0.0 (A15c) | Kazakhstan / Bt, 2015, area 1, element 1, layer 3, skeleton 6/ DA228 [31] | | | | | |
| | | Middle Ages (901–920 CE) | F2c1 | 0.0 (F2c) | Kazakhstan / Almaly, Kurgan 1, Object 1, Issyk, Tian Shan / DA126 [31] | | | | | |

(Continued)

**Table 1.** (Continued)

| Nr | Variants [17, 18]/ Gene product/ Strand | Period/Years [18] | Hg | Freq % in haplogroup | Country / Location / Sample | GB Freq. | Modern populations | Mito Tip classification | HmtVar Prediction | Mitochondrial Disease (MitoMap) |
|---|---|---|---|---|---|---|---|---|---|---|
| | | | | | **Pathogenic mutations** | | | | | |
| 9 | m.5628T>C, tRNA Alanine/L | Iron Age (400–200 BCE) | F1d | 100 (F1d) | Moldova / Glinoe / SCY308 [58] | 0.19% | Brazilian woman [59], Italy [60], Germany [61], | LP | Pathogenic (M2-P31) | CPEO / DEAF enhancer / gout (Heteroplasmy) |
| 10 | m.7543A>G tRNA Aspartic acid /H | Neolithic (8200–7700 BCE) | R2 | 3.70 (R2) | Iran / Tepe Abdul Hosein, Central Zagros / AH1 [62] | 0.09% | USA [63], Germany [64] | PP | Pathogenic (M2-P29) | MEPR (Heteroplasmy) |
| 11 | m.7554G>A, tRNA Aspartic acid/H | Neolithic (8205–7756 BCE) | R2 | 3.70 (R2) | Iran / Tepe Abdul Hosein, Central Zagros / AH2 [62] | 0.00% | UK [65] | PP | Pathogenic (M2-P40) | Myopathy, ataxia, nystagmus, migraines, lactic acidosis (Heteroplasmy) |
| | | Middle Ages (10–375 CE) | HV18 | 0.0 (H2a) | Poland / Kowalewko, Greater Poland / PCA0018 (55) | | | | | |
| 12 | m.8296A>G tRNA Lysine/H | Bronze (2872–2583 BCE) | U2d2 | 41.67 (U2d) | Russia / Grachevka, Sok River, Samara / I0371 [49] | 0.07% | Japan [66] | PP | Pathogenic (M2-P2) | DMDF / MERRF / HCM / epilepsy (Heteroplasmy) |
| 13 | m.8328G>A tRNA Lysine/H | Middle Ages (80–260 CE) | U3a1a | 0.0 (U3a) | Poland / Kowalewko, Greater Poland / PCA0054 (55) | 0.00% | UK [67] | LP | Pathogenic (M2-P39) | Mito Encephalopathy / EXIT with myopathy and ptosis (Heteroplasmy) |
| 14 | m.8342G>A rs118192103/ tRNA-Lysine/H | Middle Ages (10–375 CE) | HV18 | 0.0 (H2a) | Poland / Kowalewko, Greater Poland / PCA0018 (55) | 0.00% | Italy [68] | LP | Pathogenic (M2-P53) | CPEO and Myoclonus (Heteroplasmy) |
| 15 | m.12300G>A tRNA Leucine/H | Middle Ages (80–260 CE) | U3a1a1 | 0.0 (U3a) | Poland / Kowalewko, Greater Poland / PCA0004 (55) | 0.00% | Spain [69] | PP | Pathogenic (M0-P36) | MERRF (Heteroplasmy) |
| 16 | m.15915G>A tRNA Threonine/H | Middle Ages (80–120 CE) | T2n | 0.0 (U5b) | Poland / Kowalewko, Greater Poland / PCA0032 (55) | 0.00% | Japan [70] | PP | Pathogenic (M2-P30) | Encephalomyopathy (Heteroplasmy) |

GB Freq—GenBank frequencies; CPEO—Chronic progressive external ophthalmoplegia; MM—Mitochondrial myopathies; MERRF—Myoclonic epilepsy with ragged red fibers; MEPR—Myoclonic epilepsy and psychomotor regression; DMDF—Diabetes Mellitus + Deafness. HCM—Hypertrophic cardiomyopathy; MELAS—Mitochondrial encephalomyopathy, lactic acidosis. and stroke-like episodes; MIDD—Maternally inherited diabetes and deafness; SNHL—Sensorineural hearing loss; NARP–Neuropathy, ataxia and retinitis pigmentosa; Cfrm Path–confirmed pathogenic; LP–likely pathogenic; PP–possibly pathogenic; M—tRNA model; P–position in tRNA model (HmtVar).

## Confirmed pathogenic mutations

**m.5703G>A (rs199476130).** The variant m.5703G>A was found in five ancient mtDNA samples, one from the Neolithic period, three from the Iron Age and one from the Middle Ages. The two of the Iron Age samples are from the same site and time period in today's Russia, extracted the aged remains of a male and a female, and they probably related. This is also the pathogenic variant established in ancient mtDNA from archeological sites spanning the widest geographical range, i.e. from Poland to Mongolia. Our literature survey for the m.5703G>A mutation finds that it has been reported to cause mitochondrial myopathy (MM) and ophtalmoplegia [28, 29]. Recently, its phenotypic spectrum was broadened by a report of a patient with typical myoclonic epilepsy with ragged red fiber (MERRF) syndrome carrying a heteroplasmic m.5703G>A mutation [30]. In another recent study, however, it has been argued that investigations carried out to confirm the pathogenicity of this variant are insufficient [71].

**m.3243A>G (rs199474657).** This variant was detected in a sample from a site in Germany from the Bronze Age (2029–1911 BCE) (Table 1). Population-based studies suggest the m.3243A>G mutation is one of the most common disease-causing mtDNA mutations, with a carrier rate of 1 in 400 people [72, 73]. This mutation has been shown to be associated with a wide range of symptoms, and there is evidence the outcome is also being determined by nuclear genetic factors [34]. It is associated with mitochondrial encephalopathy, lactic acidosis and stroke-like episodes (MELAS) [74], maternally inherited deafness and diabetes (MIDD) [75] and chronic progressive external ophthalmoplegia (CPEO) [76]. Other reported features

include renal failure [77], isolated myopathy, cardiomyopathy, seizures, migraine, ataxia, cognitive impairment, bowel dysmotility and short stature [78]. Low to moderate levels of mutant heteroplasmy in the m.3243G>A mutation are often associated with MIDD, whereas higher levels are variably associated with myopathy, high frequency sensorineural hearing loss, short stature, epilepsy, strokes and dementia [11, 79, 80]. Elevated heteroplasmy levels have in general been shown to lead to neurologic, movement, metabolic, and cardiopulmonary impairments [81].

**m.5650G>A.** This variants was found in 3 samples from the Iron Age, but they are from different Central Asian sites and time periods (ranging from 900 BCE to 134 BCE), so the individuals they were taken from are in all likelihood unrelated. McFarland and colleagues (2008) report a family where proximal myopathy has become increasingly severe with successive generations of the maternal lineage, and this pure myopathy is shown to be caused by the m.5650G>A mutation [39]. Finnila et al. (2001) describe a patient with MERRF, who had a *de novo* m. 5650G>A mutation in the tRNA Alanine gene [41]. The mutation was heteroplasmic, with the proportions of the mutant genome being up to 99% in muscle. They suggest that the mtDNA mutation is pathogenic as it was associated with relevant clinical phenotype, it is absent in controls, and it alters a structurally important segment in the amino acid acceptor stem in the tRNA Alanine.

**m.8340G>A.** The variant was found in one sample from the Middle Ages (10–375 CE). A number of studies have established association between the m.8340G>A variant and various clinical conditions. Jeppesen et al. (2014) report two patients with a *de novo* m.8340G>A variant associated with exercise intolerance, CPEO and myopathy [82]. Gill et al. (2017) report a patient carrying this mutation with cataracts, pigmented retinopathy, rod-cone dysfunction and sensory neural deafness without myopathy [43]. Tamopolsky et al. (2019) report a case with a *de novo* m.8340G>A DNA mutation associated with mitochondrial myopathy, ptosis and ophthalmoparesis, corroborating the pathogenicity of the m.8340G>A mutation [42]. The collective data is consistent with a categorization of "pathogenic" given that the variant is found in much higher frequency in patients (3 reports) *vs* a control population, and with much higher heteroplasmy levels reported in ragged blue fibers and COX-negative *vs* healthy fibers [43, 82].

**m.14674T>C.** The variant was found in one sample from Poland from the Bronze Age (1592–1591 BCE). A study by Houshmand et al. (1994) describes two patients with Mitochondrial myopathy (MM) that are carriers of the m.14674T>C mutation. Nucleotides that are conserved between species are unaffected by this mutation, and the authors presume that it is unlikely to be pathogenic. This homoplasmic mutation has been identified in reversible infantile cytochrome c oxidase deficiency (or "Benign COX deficiency") [47]. Carriers of this mutation experience subacute onset of profound hypotonia, feeding difficulties and lactic acidosis within the first months of life. Although recovery may occur, mild myopathy persists into adulthood [45].

**m.7510T>C (rs199474820).** This variants was found in one sample from present day Bulgaria from the Neolithic period, estimated to be from 5800–5400 BCE, and is thus the oldest pathogenic variant detected in this study. A number of studies have established association between the m.7510T>C mutation and non-syndromic sensorineural hearing impairment (SNHL) [50, 51, 54, 55]. Review of the published cases suggests that there is interfamilial variability in the age of onset, accompanying symptoms, and haplogroup background [54]. The results of Kytövuori and colleagues (2017) suggest that in addition to sensorineural hearing impairment, the m.7510T>C mutation is associated with a spectrum of mitochondrial disease clinical features including migraine, epilepsy, cognitive impairment, ataxia, and tremor, and with evidence of mitochondrial myopathy [53].

### Likely/possibly pathogenic mutations

This group includes ten mutations for which there is discrepancy in their clinical effect designation between the two used platforms. Whereas HmtVar classifies them as pathogenic, Mito-Map classifies them as likely or possibly pathogenic (Table 1).

## Discussion

This study establishes for the first time the presence of pathogenic mtDNA variants in 1443 ancient mtDNA Eurasian genomes from different periods, publicly available in the AmtDB database. Among the 3191 unique variants detected in the analyzed samples, six are "confirmed pathogenic" with well-established clinical association in contemporary patients. Our results suggest that the prevalence of pathogenic mtDNA mutations might have been far greater in ancient populations, eleven cases of six unique confirmed pathogenic mtDNA mutations were established in 1443 ancient samples, compared to 1.5–2.9 in 100000 in contemporary populations [72]. This marked drop in prevalence might be related to the prevalence of particular mitochondrial haplogroups as there is evidence that certain pathogenic mutations are associated with particular haplogroup background [83].

The m.5703G>A mutation is established in European and Asian ancient samples, whereas in contemporary patients it is detected in a 16-year-old Caucasian girl in Europe and a Chinese girl, along with an African-American woman (cf. Table 1, Fig 1). Contemporary geographic distribution for this variant thus generally corresponds to that in the examined past periods. This variant was established in samples assigned to five different haplogroups (cf. Table 1),

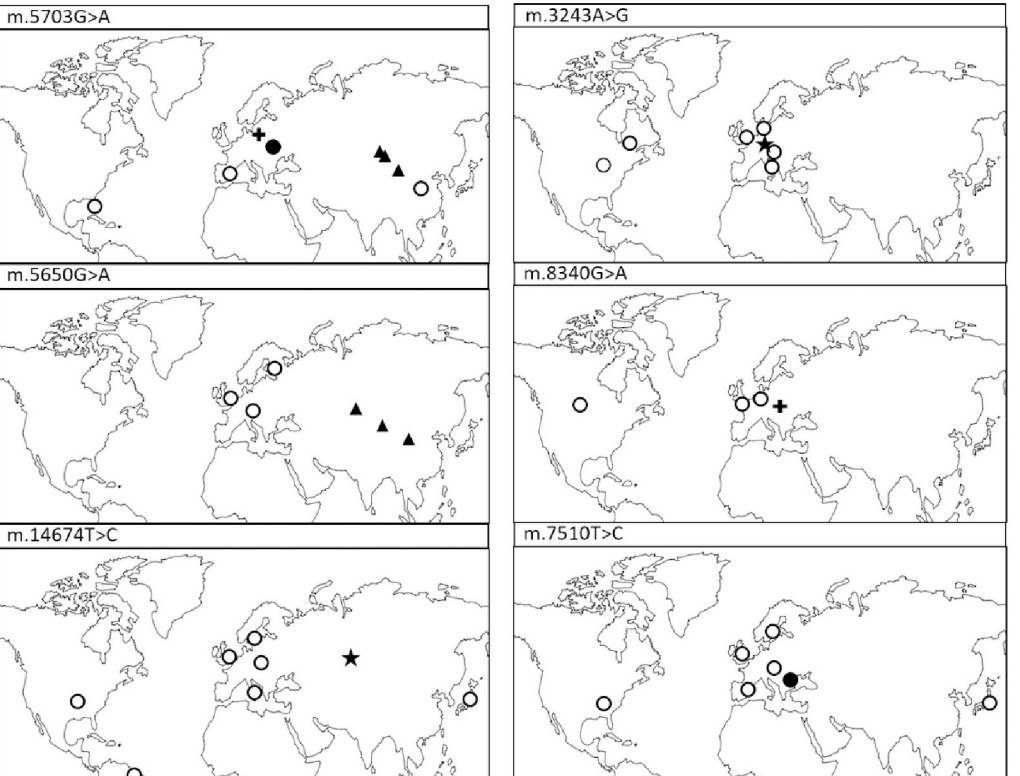

**Fig 1. Geographic location of the Eurasian archaeological sites where mtDNA samples containing "confirmed pathogenic" variants had been found (cf. Table 1).**

implying that it does not associate with a particular haplogroup, and has probably recurred in populations with different haplogroup background.

The variant m.5650G>A is established in three ancient samples from the Asian region (today's Russia, Kazakhstan and Mongolia), assigned to three different haplogroups, while contemporary patients are all in Europe (cf. Table 1, Fig 1). The sample found in today's Russia is W1c haplogroup which is determined to have arisen 8 kya [84], and is presently found in Europe, the Near East, Caucasus and India [85]. The samples found in today's Kazakhstan and Mongolia belong to East Asian haplogroups F1b1 and D4b2b2b, respectively [86]. The estimated distribution of m.5650G>A in ancient times thus generally corresponds to the contemporary distribution of the haplogroups harboring these variant. The incidence of the variant in contemporary European populations, where F1b1 and D4b2b2b lineages are rare, indicates lack of association of the variant with these haplogroups. The presence of m.5650G>A in contemporary European samples, and its absence from ancient European samples, might be the result of *de novo* mutation or migration and subsequent evolutionary events.

The remaining four ancient pathogenic variants are established in single samples belonging to different mitochondrial haplogroups. The m.14674T>C variant was determined in only one ancient sample from the territory of present day Mongolia, but contemporary carriers of this variant are found in the U.S., Germany, Brazil, Italy, Japan, the UK and Sweden. This ancient sample is classified to haplogroup U2e1h, a sub-branch of haplogroup U2e, presently found at low frequencies in Europe and Western Asia [87]. The correspondence of the contemporary distribution of m.14674T>C with the geographic range of the relatively rare U2e1h lineage indicates possible association between this variant and haplogroup.

Overlap between the geographical location of the ancient sample and the present-day distribution of the corresponding mitochondrial haplogroup is found in three more pathogenic mutations: m.8340G>A established in an ancient sample from present day Poland (assigned to the Western Eurasian haplogroup HV18); m.7510T>C detected in an ancient sample from present day Bulgaria (designated to haplogroup J2b1, found among modern populations in Atlantic Europe, the East European Plain and the Near East); and m.3243A>G found in ancient sample from Germany (belonging to a subclade of T2f, considered almost exclusively European, with rare instances in the Near East) [88].

The established "confirmed pathogenic" mutations are detected in samples estimated to be between 1600 and 7800 years old. The oldest of these, m.7510T>C, is detected in a Neolithic sample (5800–5400 BCE). The m.5703G>A mutation was detected in a Neolithic sample that could be as old as 4800 years, but it is found again in samples from later periods, Iron Age (900–600 BCE) and Middle Age (10–375 CE). Other mutations with established long histories, detected in >4000 years old Bronze Age samples are m.3243A>G, the most common mtDNA mutation with pathogenic effect, present in contemporary populations with 0.02% frequency, and m.14674T>C, found in a 2600 old Bronze Age sample, and also found in contemporary populations with 0.01% frequency (cf. Table 1). The m.5650G>A mutation is detected in an Iron Age sample that could be as old as 2900 years. The low population frequency of these variants in contemporary populations is further corroboration of their pathogenic significance.

Two mutations with likely/possibly pathogenic effect, m.7543A>G and m.7554G>A, detected in Neolithic period samples from the Neolithic period from the same archeological site and time span, are the oldest established mutations with putatively pathogenic effect (10200–10000 years old). Also, two Bronze Age mutations, m.8296A>G and m.4440G>A, could be as old as 4800 years. It is noteworthy that m.4440G>A is detected in as many as 12 ancient samples from different time periods and locations. Its pathogenic effect is corroborated by that, despite it being the most common clinically significant mutation detected in ancient

mtDNA samples in this study, it has been described as a novel mutation causing MM in contemporary patients [57].

It is noticeable that all the established ancient mutations putatively associated with diseases are located in tRNA genes, and none is in genes encoding the 13 essential polypeptides of the OXPHOS system, even though tRNAs comprise only about 10% of the total coding capacity of the mitochondrial genome [89]. Epidemiological studies have highlighted that point mutations in the mt-tRNA genes are among the most common defects observed [73, 90]. Mitochondrial tRNA mutations have been shown to be the most prevalent genetic defect by a survey of an adult population with mtDNA disease, accounting for more than 50% of all genetically diagnosed cases [91]. More than 150 different point mutations have been described in mt-tRNA genes including novel disease-causing mutations. Associated pathogenic mechanisms continue to be identified [92], yet mtRNA mutations'role in interfering with the translation mechanism remains unclear.

Ancient mutations putatively associated with mitochondrial diseases are in different tRNA genes and affect nucleotides in different functional parts of the encoded tRNA molecule (Table 2).

The gene *MT-TK* coding tRNA Lysine is most commonly affected by mutations putatively associated with mitochondrial diseases in ancient mtDNA samples, i.e. one confirmed pathogenic (m.8340G>A) and three likely/possibly pathogenic (m.8296A>G, m.8328G>A and m.8342G>A). It is conceivable that these mutations have different clinical significance related to their impact on the stability of the tRNA protein.

A recent study on another mtDNA mutation in the same gene - m.8344 A>G, known to be associated with MERRF, has demonstrated that tRNA modifications have distinct effects on the stability and synthesis of mitochondrial proteins [82]. Such regulating mechanisms might play role in the etiology of human disease, and new RNA sequencing approaches to mitochondria should provide insights. Two ancient mutations were established in the *MT-TD* coding Aspartic acid. Aspartic acid is neurotransmitter and recent studies show that it may be

**Table 2. Nucleotide position in tRNA cloverleaf secondary domains of ancient mutations in tRNAs.**

| Nr | Mutation | Gene | Nucleotide position in tRNA cloverleaf secondary domains (loops and stems) |
|---|---|---|---|
| | | **Confirmed Pathogenic** | |
| 1 | m.5703G>A | MT-TN | Position 27 in CS—Anticodon Stem |
| 2 | m.3243A>G | MT-TL1 | Position 14 in DL—Dihydrouridine Loop |
| 3 | m.5650G>A | MT-TA | Position 6 in AS—Acceptor Stem |
| 4 | m.8340G>A | MT-TK | Position 51 in TS—TψC Stem |
| 5 | m.14674T>C | MT-TE | Position 73 in E—3' End |
| 6 | m.7510T>C | MT-TS1 | Position 5 in AS—Acceptor Stem |
| | | **Likely/Possibly Pathogenic** | |
| 7 | m.1624C>T | MT-TV | Position 25 in DS—Dihydrouridine Stem |
| 8 | m.4440G>A | MT-TM | Position 42 in CS—Anticodon Stem |
| 9 | m.5628T>C | MT-TA | Position 31 in CS—Anticodon Stem |
| 10 | m.7543A>G | MT-TD | Position 29 in CS—Anticodon Stem |
| 11 | m.7554G>A | MT-TD | Position 40 in CS—Anticodon Stem |
| 12 | m.8296A>G | MT-TK | Position 2 in AS—Acceptor Stem |
| 13 | m.8328G>A | MT-TK | Position 39 in CS—Anticodon Stem |
| 14 | m.8342G>A | MT-TK | Position 53 in TS—TψC Stem |
| 15 | m.12300G>A | MT-TL2 | Position 36 in CL—Anticodon Loop |
| 16 | m.15915G>A | MT-TT | Position 30 in CS—Anticodon Stem |

involved in the pathogenesis of a stroke-like episode [43]. The remaining putatively pathogenic mutations established in ancient mtDNA are located in different tRNA genes.

It is notable that the established ancient mutations are located in different cloverleaf models of tRNAs. (Fig 2). The tRNA model indicates one of the four possible groups human mt-tRNAs are classified based on their structural diversity and tertiary interactions [93]. Model 0 represents the quasi-canonical cloverleaf structure, with standard D-loop/T-loop interaction; Model 1- a single tRNA with an atypical anticodon stem; Model 2 –the most common among mt-RNAs, is characterized by loss of D-loop/T-loop interaction and Model 3—lack of D-stem. Thirteen out of the sixteen mutations putatively associated with mitochondrial disease presented in this study are in Model 2.

Seven out of the 16 mutations considered here are located in CS-Anticodon Stem, four in AS-Acceptor Stem, two in TS-TΨC Stem, and single mutations are located in DL-Dihydrouridine Loop, CL-Anticodon Loop and DS-Dihydrouridine Stem (Fig 2).

Confirmed pathogenic mutation m.5703G>A is located on position 27 in Model 2 tRNA which is involved in post-transcriptional modifications and is adjacent to position 26, which participates in tertiary folding and is also subject to posttranscriptional modifications. Confirmed pathogenic mutation m.3243A>G in Model 0 t RNA affects position 14, which is involved in tertiary folding with interactions represented by lines on Fig 2. Mutation m.14674T>C in Model 2 tRNA is in position 73 in 3' End of the acceptor stem, which participates in 3' end-editing in the final stages of tRNA formation. The role of the remaining three confirmed pathogenic mutations is more difficult to be determined. Mutation m.5650G>A is in position 6 of the acceptor stem of a Model 2 tRNA. Pathogenic mutation m.8340G>A in position 51 of a Model 2 tRNA is located next to position 50 nucleotide involved in post-transcriptional modifications.

Three mutations with putatively pathogenic effect are in positions that affect the structure and the function of the tRNA, m.12300G>A in position 36 of the CL-Anticodon Loop, m.8328G>A in position 39 and m.7554G>A in position 40 that all have impact on post-transcriptional modifications.

Mutations in tRNA genes could lead to disturbances of three dimensional structure, the absence of post-transcriptional modifications of the tRNA, rise in the number of errors and thus to tRNA destabilization. This could result in accumulation of cell damaging proteins and

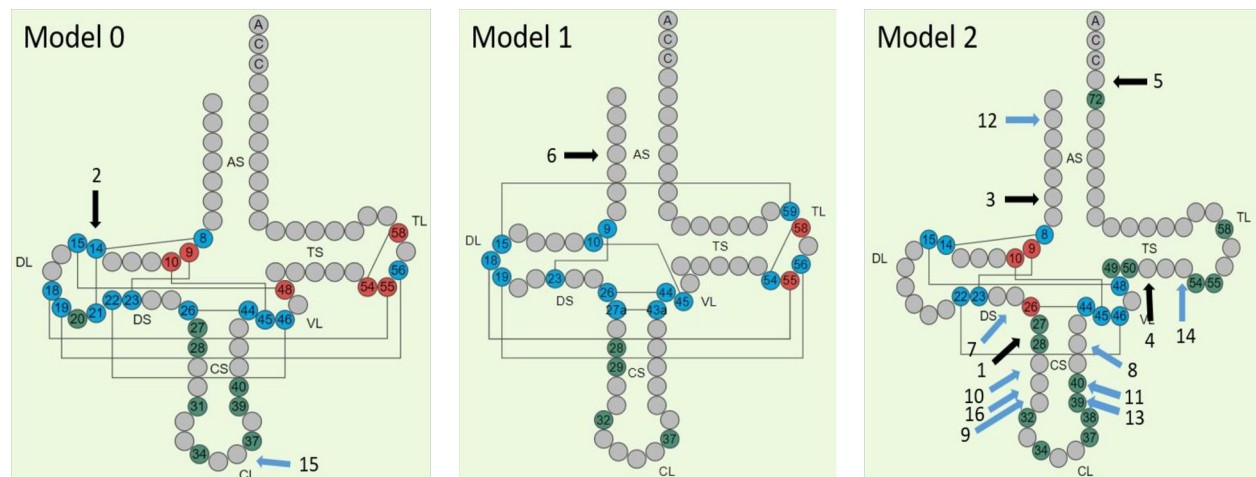

**Fig 2. Location on tRNA nucleotide sequence of ancient Cfrm/LP/PP mutations (HmtVar).** The positions of the tRNA nucleotide sequence affected by mutations are indicated with a black arrows for cfrm mutations and blue arrows for LP/PP mutations.

cutback of mitochondrial protein synthesis, including reduction of OXPHOS proteins. Despite the clinical significance, the molecular mechanisms leading to such disturbances remain poorly understood.

As often happens in ancient DNA analyses, because the amount of endogenous template DNA is typically very low, the surviving molecules are typically short and affected by post-mortem cytosine deamination damage which appears as C>T and G>A variants in sequence data [94]. It is noteworthy to mention that 12 out of the 19 (63.2%) pathogenic or putatively pathogenic mutations that we detect in the analyzed ancient mtDNA samples are G>A or C>T substitutions. Review of the publications that present the analyzed ancient mtDNA genomes substantiates that the authors have employed adequate analyses to mitigate the effect of post mortem damage (PMD), strengthening our confidence that these are real variants and not the result of PMD [27, 31, 33].

Still, pathogenic mutations in mitochondrial DNA often show highly variable phenotypes for any given point mutation, and severity of the clinical and biochemical phenotype has been roughly proportionate to the percent mutant heteroplasmy [10, 79]. Identifying heteroplasmic variants and establishing the level of heteroplasmy in ancient samples is not a trivial task. Heteroplasmic variants however constitute the bulk of disease-associated mtDNA variants in contemporary humans, and most of the detected confirmed or putatively pathogenic variants in our study have pathogenic effect in heteroplasmic state. Nevertheless due to insufficient phenotypic data about the human remains, there is no way of exactly knowing if disease-associated mutations, or those predicted to have a strong functional effect, were indeed pathogenic in ancient populations.

## Conclusion

The established mtDNA pathogenic mutations in the analyzed ancient samples, all in tRNA coding genes, are putatively associated with a wide range of mitochondrial diseases found in contemporary populations. Studying putative pathogenic mutations from ancient mtDNA informs on the mitochondrial disease spectrum in ancient times, and comparing their frequencies among populations separated by significant time periods sheds light on the history of the disease. Our findings suggest that disease associated genes are often genes with long history, and that pathogenic variants in mtDNA exhibit diverse temporal dynamics with regard to their geographic distribution and mitochondrial haplogroup background association. Exposing pathogenic variants in ancient human populations contributes to our understanding of their origin and prevalence dynamics.

## Author Contributions

**Conceptualization:** Draga Toncheva.

**Methodology:** Draga Toncheva, Dimitar Serbezov, Sena Karachanak-Yankova, Desislava Nesheva.

**Supervision:** Draga Toncheva.

**Validation:** Dimitar Serbezov, Sena Karachanak-Yankova, Desislava Nesheva.

**Visualization:** Dimitar Serbezov.

**Writing – original draft:** Draga Toncheva, Dimitar Serbezov, Sena Karachanak-Yankova, Desislava Nesheva.

**Writing – review & editing:** Draga Toncheva, Dimitar Serbezov, Sena Karachanak-Yankova, Desislava Nesheva.

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
