## [Decision Letter · Decision Letter 0]

23 Jun 2020

PONE-D-20-13726

Ancient mitochondrial DNA pathogenic variants putatively associated with mitochondrial disease

PLOS ONE

Dear Dr. Toncheva,

Thank you for submitting your manuscript to PLOS ONE. After careful consideration, we feel that it has merit but does not fully meet PLOS ONE’s publication criteria as it currently stands. Therefore, we invite you to submit a revised version of the manuscript that addresses the points raised during the review process.

We look forward to receiving your revised manuscript.

Kind regards,

David Caramelli, Ph.D

Academic Editor

PLOS ONE

Journal Requirements:

Additional Editor Comments (if provided):

Reviewers' comments:

Reviewer's Responses to Questions

**Comments to the Author**

1. Is the manuscript technically sound, and do the data support the conclusions?

Reviewer #1: Yes

2. Has the statistical analysis been performed appropriately and rigorously? 

Reviewer #1: N/A

3. Have the authors made all data underlying the findings in their manuscript fully available?

Reviewer #1: Yes

4. Is the manuscript presented in an intelligible fashion and written in standard English?

Reviewer #1: Yes

5. Review Comments to the Author

Reviewer #1: In the present paper, authors investigated on pathogenic mutations in ancient mtDNA, providing further insight into the emergence of mitochondrial diseases.

Although not all available ancient mitogenomes were considered, sources were reported and the analyses were detailed and appropriate. I have minor concerns regarding this study as it is currently presented and discussed. The authors could make modifications throughout the text to increase the quality of the manuscript and to comply with the standards for publication in Plos One. I suggest authors to improve their discussion by exploring the distribution of the mitochondrial pathogenic variants in modern populations, their geographic distribution and/or their differential distribution in the mitochondrial haplogroups. The latter could be an interesting discussion if compared with the mitochondrial haplogroup found in ancient samples harboring the same pathogenic mutations.

Few minor points to be checked are the following:

Line 44 “Exposing pathogenic variants in ancient human populations expands our understanding of their origin”: Authors could better argue this sentence in the conclusion paragraph and further explain the importance of this study.

Line 115 “confirmed pathogenic|” should modified in “confirmed pathogenic”

Line 135 “mitochondrial myopathy and ophtalmoplegia (MM)” MM is for mitochondrial myopathy thus I suggest to modify as follows: “mitochondrial myopathy (MM) and ophtalmoplegia”.

Line 144 “Population-based studies suggest the m.3243A>G mutation is the most common disease-causing…” Sentence has something wrong, please verify and change.

Line 88: “We used various publicly available databases to gather information to identify mtDNA sequence variants and to gather information about each variant…” Sentence has too much repetitions, please rephrase.

Line 161 I would prefer McFarland and colleagues instead of McFarland et al.

Line 175 Gill et al. Instead of Gill et al

The same for Houshmand et al line 187, and line 202 where “et al” should be “et al.“

Line 188-190. Sentence is confusing. Please rephrase

Table 1: notes are unclear because they are covered by row numbering. If I am right MERRF and

MEPR acronyms are reported twice.

Authors consider samples dated before 375CE as Middle Ages as reported in AmtDB database. As they belong to the Roman period, I think that this historical classification is arguable.

Line 206-209: “Likely/possibly pathogenic mutations” I suggest authors to briefly describe in this paragraph the main results even if they are summarized in the table.

Line 275: “Model 0 model” the word model is present twice.

Line 365 “…mutation in the mitochondrial tRNA <sup>Leu(UUR)</sup> gene. Diabetes. 1994;43(6):746”. Please change in “…mutation in the mitochondrial tRNA(Leu(UUR)) gene. Diabetes. 1994;43(6):746‐751”.

6. PLOS authors have the option to publish the peer review history of their article (what does this mean?). If published, this will include your full peer review and any attached files.

Reviewer #1: No

---

## [Author Response · Author response to Decision Letter 0]

30 Jul 2020

Dear Sirs,

Thank you for reviewing our manuscript and for letting us resubmit a revised version. We find all comments to be very helpful and constructive. We have taken up your suggestions, and the manuscript has been thoroughly revised. Notably, following the advice of the reviewer, we now discuss the spatiotemporal dynamics of the putatively pathogenic variants established in ancient samples based on their geographic location, contemporary epidemiological trends and haplogroup affiliation. Below, we explain in detail how we have addressed the various comments and suggestions. We believe that these changes have resulted in a much better version of the manuscript, which we hope is more befitting for publication in PLOS ONE.

Sincerely,

Prof. Draga Toncheva

Academic editor`s comments:

Our Reply: We have acquainted ourselves with PLOS ONE style templates and made sure the manuscript meets the journal`s style requirements

2. PLOS requires an ORCID iD for the corresponding author in Editorial Manager on papers submitted after December 6th, 2016. Please ensure that you have an ORCID iD and that it is validated in Editorial Manager.

Our Reply: We updated the corresponding author`s information in Editorial Manager and validated the ORCID iD.

Our reply: We have now replaced this with our own very basic image that we deem sufficient.

REVIEWER(S)' COMMENTS:

Reviewer #1: In the present paper, authors investigated on pathogenic mutations in ancient mtDNA, providing further insight into the emergence of mitochondrial diseases.

Although not all available ancient mitogenomes were considered, sources were reported and the analyses were detailed and appropriate. I have minor concerns regarding this study as it is currently presented and discussed.

The authors could make modifications throughout the text to increase the quality of the manuscript and to comply with the standards for publication in Plos One.

Our Reply: We have thoroughly revised our manuscript to meet the standards for publication in PLOS ONE.

I suggest authors to improve their discussion by exploring the distribution of the mitochondrial pathogenic variants in modern populations, their geographic distribution and/or their differential distribution in the mitochondrial haplogroups. The latter could be an interesting discussion if compared with the mitochondrial haplogroup found in ancient samples harboring the same pathogenic mutations.

Our Reply: We have now added additional text at the beginning of the Discussion section exploring the overlap in distribution of the pathogenic variants in ancient and modern populations, and their differential distribution in the mitochondrial haplogroups.

Few minor points to be checked are the following:

Line 44 “Exposing pathogenic variants in ancient human populations expands our understanding of their origin”: Authors could better argue this sentence in the conclusion paragraph and further explain the importance of this study.

Our Reply: We have now added similar sentence in the “Conclusion” paragraph

Line 115 “confirmed pathogenic|” should modified in “confirmed pathogenic”

Our Reply: We have corrected this.

Line 135 “mitochondrial myopathy and ophtalmoplegia (MM)” MM is for mitochondrial myopathy thus I suggest to modify as follows: “mitochondrial myopathy (MM) and ophtalmoplegia”.

Our Reply: We have corrected this.

Line 144 “Population-based studies suggest the m.3243A>G mutation is the most common disease-causing…” Sentence has something wrong, please verify and change.

Our Reply: The phrase “the most common disease-causing” is changed to “one of the most common disease-causing”, and a reference is added to substantiate this claim.

Line 88: “We used various publicly available databases to gather information to identify mtDNA sequence variants and to gather information about each variant…” Sentence has too much repetitions, please rephrase.

Our Reply: This sentence has now been changed to “We used various publicly available databases to collect information on mtDNA sequence variants, including its clinical significance and contemporary population frequencies”

Line 161 I would prefer McFarland and colleagues instead of McFarland et al.

Our Reply: We have now changed that.

Line 175 Gill et al. Instead of Gill et al

The same for Houshmand et al line 187, and line 202 where “et al” should be “et al.“

Our Reply: We have now corrected these.

Line 188-190. Sentence is confusing. Please rephrase

Our Reply: You are absolutely right! We have now changed this sentence to “Nucleotides that are conserved between species are unaffected by this mutation, and the authors presume that it is unlikely to be pathogenic.”

Table 1: notes are unclear because they are covered by row numbering. If I am right MERRF and

MEPR acronyms are reported twice.

Our Reply: We have now corrected this.

Authors consider samples dated before 375CE as Middle Ages as reported in AmtDB database. As they belong to the Roman period, I think that this historical classification is arguable.

Our Reply: We have used the annotation given in the Ancient mtDNA database (Ehler E, et al. AmtDB: a database of ancient human mitochondrial genomes. Nucleic acids research. 2018;47), where these samples were taken from.

Line 206-209: “Likely/possibly pathogenic mutations” I suggest authors to briefly describe in this paragraph the main results even if they are summarized in the table.

Our Reply: We have now added that they are ten mutations. We choose to concentrate our attention to the group of “confirmed pathogenic” mutations and feel additional mention of these results is unnecessary.

Line 275: “Model 0 model” the word model is present twice.

Our Reply: We have now corrected this.

Line 365 “…mutation in the mitochondrial tRNA ^Leu(UUR)^ gene. Diabetes. 1994;43(6):746”. Please change in “…mutation in the mitochondrial tRNA(Leu(UUR)) gene. Diabetes. 1994;43(6):746‐751”.

Our Reply: We have now corrected this.

---

## [Editor Report · Decision Letter 1]

11 Aug 2020

Ancient mitochondrial DNA pathogenic variants putatively associated with mitochondrial disease

PONE-D-20-13726R1

Dear Dr. Toncheva,

We’re pleased to inform you that your manuscript has been judged scientifically suitable for publication and will be formally accepted for publication once it meets all outstanding technical requirements.

Kind regards,

David Caramelli, Ph.D

Academic Editor

PLOS ONE
---

## [Editor Report · Acceptance letter]

9 Sep 2020

PONE-D-20-13726R1 

Ancient mitochondrial DNA pathogenic variants putatively associated with mitochondrial disease 

Dear Dr. Toncheva:

I'm pleased to inform you that your manuscript has been deemed suitable for publication in PLOS ONE. Congratulations! Your manuscript is now with our production department. 

Kind regards, 

on behalf of

Professor David Caramelli 

Academic Editor

PLOS ONE